

# Hurricane eyewall winds and structural response of wind turbines

Amber Kapoor[1], Slimane Ouakka[2], Sanjay R. Arwade[2], Julie K. Lundquist[3,4], Matthew A. Lackner[1], Andrew T. Myers[5], Rochelle P. Worsnop[6,7], George H. Bryan[8]

[1]Mechanical & Industrial Engineering, University of Massachusetts Amherst, Amherst, MA 01003 USA
[2]Civil & Environmental Engineering, University of Massachusetts Amherst, Amherst, MA 01003 USA
[3]Atmospheric & Oceanic Sciences, University of Colorado Boulder, Boulder, CO 80309 USA
[4]National Renewable Energy Laboratory, Golden, CO 80401 USA
[5]Civil & Environmental Engineering, Northeastern University, Boston, MA 02115 USA
[6]University of Colorado, Cooperative Institute for Research in Environmental Sciences, Boulder, CO, UDA
[7]NOAA/Earth System Research Laboratory, Physical Sciences Division, Boulder, CO 80305 USA
[8]National Center for Atmospheric Research, Boulder, CO 80305 USA

*Correspondence to:* Sanjay R. Arwade (arwade@umass.edu)

**Abstract.**

This paper describes the analysis of a wind turbine and support structure subject to simulated hurricane wind fields. The hurricane wind fields, which result from a Large Eddy Simulation of a hurricane, exhibit features such as very high gust factors (>1.7), rapid direction changes (30° in 30 sec) and substantial veer. Wind fields including these features have not previously been used in an analysis of a wind turbine and their effect on structural loads may be an important driver of enhanced design considerations. With a focus on blade root loads and tower base loads, the simulations show that these features of hurricane wind fields can lead to loads that are substantially in excess of those that would be predicted were wind fields with equally high mean wind speeds but without associated direction change and veer used in the analysis. This result, if further verified for a range of hurricane and tropical storm simulations, should provide an impetus for revisiting design standards.

## 1 Introduction

Activity related to offshore wind energy development continues to accelerate along the US East Coast. For example, the Massachusetts legislature recently proposed to double Massachusetts' offshore wind energy commitment to 3,200 MW (Massachusetts HR 2018), and a public price was tentatively set for the first 800 MW of offshore wind energy capacity in Massachusetts, an important step in the process of developing offshore wind farms. Awareness of potential hurricane risk to offshore wind farms along the US East Coast is high, and with potentially thousands of offshore wind turbines to be constructed over the coming decades, quantification of this risk is crucial to ensure the safety, availability, and reliability of this important energy source.

Analytic models of hurricane wind fields (e.g. (Holland, 1980); (Holland et al., 2010)), based on interpolations between sparse observations, have been useful for informing offshore wind design specifications (Tarp-Johansen and Clausen 2006; Hallowell et al. 2018). However, these models cannot capture certain non-stationary features of



hurricane eyewalls. Recent advances in observations ((Stern et al., 2016); (Wingo and Knupp, 2016); (Wurman and Kosiba, 2018)) and in hurricane simulation capabilities (Worsnop et al. 2017a; Wu et al. 2018a) allow identification

of sub-km scale features of hurricane wind fields, e.g., organized turbulent structures such as mesovortices, that may create significant and unforeseen loads on offshore wind turbines. Many of these characteristics are extremely difficult to measure during actual storms because they occur in or near the eyewall of the hurricane where wind speeds are extremely high and observations are difficult to make. Observation of the distribution and lifetime of these mesovortices also may require simultaneous measurement of wind speed and direction over spatial domains measured

in the tens of kilometers.

The limited set of observations can be augmented through the use of Large-Eddy Simulations (LES) that resolve these turbulent structures. LES of hurricane wind fields can provide insight into organized structures within the hurricane eyewall, such as convective cells and transient large eddies. By using LES, the energy-producing scales of three-dimensional atmospheric turbulence should be explicitly resolved, while finer-scales of turbulence are parameterized

by a sub-grid-scale model. Reducing the grid size of an LES allows more of the three-dimensional turbulence to be resolved (i.e., computed directly) rather than parameterized through the subgridscale turbulence model. As a result, the maximum instantaneous wind speed in organized structures in the eyewall produced in these LES tends to be highly sensitive to the resolution of the simulation (Rotunno et al., 2009).

The LES of (Zhu, 2008) investigated hurricane dynamics driven by realistic mesoscale weather forcing, with a finest

horizontal resolution of 100 m in the innermost domain. This domain only comprised a small region of the inner core of the hurricane. The idealized simulations of (Rotunno et al., 2009), at 62-m horizontal resolution, indicated very strong mesovortices with maximum instantaneous wind speeds of 120 m s$^{-1}$. (Green and Zhang, 2015) explored how model resolution and the representation of the boundary-layer affects the development of mesovortices and other fine-scale structures in the hurricane boundary layer, with some simulations as fine as 111-m horizontal resolution. They

also suggest that the resolution of their simulations affected the size of the large-eddy circulations, implying that finer resolution was required for convergence to "true LES." (Worsnop et al., 2017a) simulated an idealized Category 5 hurricane with 32-m horizontal resolution using the Cloud Model I CM1 model of (Bryan et al., 2016). Similar CM1 hurricane simulations with 62-m horizontal grid spacing were validated with observations of turbulence spectra by (Worsnop et al., 2017b). Finally, the 37-m nested LES of Wu et al. (2018a) replicate tornadic structures within the

eyewall of Typhoon Matsa.

Since industrial-scale offshore wind energy development along the US East Coast began to be discussed seriously, a series of studies have sought to quantify the degree of risk posed to offshore wind farms by hurricanes. These studies have included attempts to identify appropriate structural performance levels and nonlinear structural analysis methods for offshore wind structures (Wei et al., 2014, 2016), multihazard risk analyses (Hallowell et al., 2018; Kim and

Manuel, 2016; Mardfekri and Gardoni, 2015; Valamanesh et al., 2015, 2016), and analysis of wind-structure interaction (Amirinia and Jung, 2017). On the whole, these studies have shown that hurricane winds can indeed pose important risks to offshore wind turbines, but that such risk can be mitigated by appropriate design approaches. None



of these studies, however, have incorporated the kind of high-resolution characterizations of hurricane wind fields that were described by (Worsnop et al., 2017a). Therefore, while a substantial body of literature already exists related to
the overall exposure of US East Coast offshore wind farms to hurricanes, this paper advances the state of knowledge by assessing the impact of specific and intense hurricane wind field characteristics that have not yet been considered.

Here, we analyze winds and turbulence from a LES of an idealized Category 5 hurricane using Cloud Model 1 (CM1), a three-dimensional, non-hydrostatic, non-linear, time-dependent numerical model designed for idealized studies of atmospheric phenomena (Bryan and Rotunno, 2009a). These simulations have been used to identify wind field
characteristics such as gust factors, spatial coherence, velocity spectrum, shear profile, direction change and veer (Worsnop et al., 2017a) that may be important drivers of offshore wind turbine response. We expand upon previous work by providing these wind characteristics to the wind field simulator TurbSim (Jonkman and Buhl Jr., 2009), and evaluating TurbSim's performance against LES. TurbSim generates simulated wind fields representative of several of the characteristics identified in the LES simulations. Finally, the effects of this flow on a wind turbine are simulated
using the DTU 10-MW reference wind turbine (Bak et al., 2013), which is a three-bladed, upwind, variable speed turbine with a rotor diameter of 178 m and a 119-m hub height. This turbine is represented in FAST, a fully coupled aero-hydro-servo-elastic code developed by the National Renewable Energy Laboratory (Jonkman and Buhl 2005). (Sim et al., 2012) used a similar procedure of defining FAST input by LES to establish guidelines on the spatial and temporal resolution needed to ensure accurate estimates of wind turbine response to turbulent wind fields. Their
simulations used the NREL 5-MW reference turbine and did not consider hurricane wind fields.

The remainder of the paper is organized into three sections. First, characteristics of the LES hurricane wind fields are described in Sect. 2. Next, simulation of representative wind fields using TURBSIM is described in Sect. 3. Finally, results of the FAST analysis are introduced to quantify the effect of hurricane wind fields on blade and tower structural demands in Sect. 4.

**2 Hurricane wind field characteristics**

This section describes a statistical characterization and analysis of a Large-Eddy Simulation (LES) of an idealized Category 5 hurricane using Cloud Model 1 (CM1) (Bryan and Rotunno, 2009b), described in detail in (Worsnop et al., 2017a). The characteristics of the simulated hurricane are based on Hurricane Felix, which made landfall in southern Mexico after traveling westward across the southern Caribbean in 2007. Although some of the
characteristics of the hurricane were derived from Hurricane Felix, the results can be taken as representative of relatively small Category 5 storms. The outer simulation domain of 3000 km x 3000 km x 25 km includes the complete hurricane, including the eyewall and rainbands. To resolve turbulent motions in the eye and eyewall, an inner fine-mesh domain of 80 km x 80 km x 3 km uses horizontal grid spacing of 31.25 m and a vertical grid spacing of 15.625 m. The model timestep is 0.1875 s. Four hours after initialization, a steady-state is achieved, and
a subsequent ten minutes of output is archived. For simplicity, the hurricane is specified to have zero translational velocity. In this paper, a subset of the complete simulation data is analyzed that covers a domain of 60 km x 60 km x

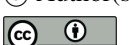



508 m using virtual towers with horizontal spacing of 1 km x 1 km. This domain encompasses the eyewall (the most intense part of the storm) and provides sufficient resolution to characterize features of hurricane wind fields that are most likely to pose a danger to offshore wind energy installations.

The features of the LES wind field are analyzed in the rest of this section, with the goal of providing representative characterizations of the wind field for subsequent use in wind turbine simulations. The wind field statistics and features quantified over the 10-minute reference period are wind speed time history statistics up to fourth order, 3-sec gusts and gust factors, wind shear profile, 10-sec and 30-sec wind direction change, and veer (change in wind direction with respect to height).

**2.1 Overall wind speeds and selection of included grid points**

The simulated hurricane includes the quiescent eye, the turbulent eyewalls, and the outer rainbands as seen in the averaged horizontal wind speed over the full 10-min simulation at an elevation (117.19 m) closest to the DTU turbine hub height of 119 m (Fig. 1). The eye of the hurricane, within which the wind speeds are low, extends to a radius $R$ of approximately 10 km. The maximum mean wind speed of approximately 90 m/s occurs at a radius of approximately

12 km and then decreases as the distance from the eye increases. For subsequent analysis and input for turbine simulations, the LES wind fields are characterized at discrete radii from the hurricane center: 10 km (inner edge of eyewall for this hurricane); 12 km (the radius of maximum mean wind speed for this hurricane); 15 km (approximate outer edge of the eyewall for this hurricane), and 20 km (well outside of the eyewall for this hurricane).

For each discrete reference radius, multiple points in the LES domain are considered to increase the sample size for

generating relevant statistics. Any grid point within +/- 100 m of a reference radius is considered to be associated with that radius (Fig. 2); a larger envelope could incorporate too large of a range of changing wind fields, especially within the eyewall. This approach yields 28 points at radii of 10 km and 12 km, 20 points at 15 km, and 40 points at 20 km. Points of interest include the hub height and the bottom and top of the rotor disk, or more specifically, the grid point elevations closest to those elevations for the DTU turbine (117.19 m, 39.06 m, 210.94 m, respectively).





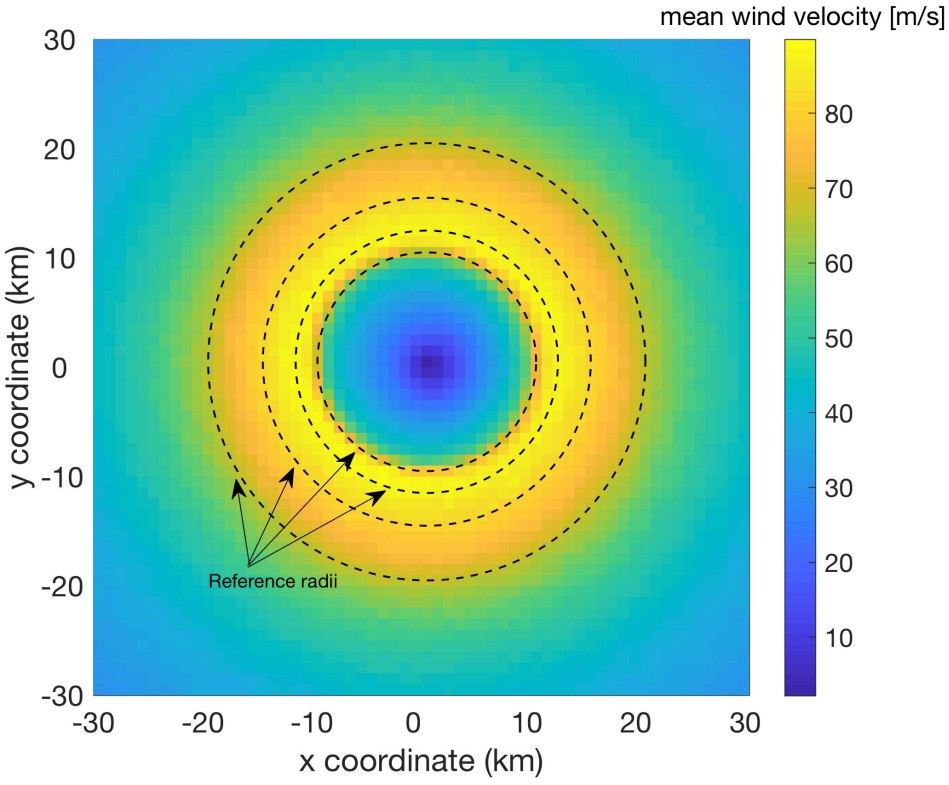

**Figure 1: Mean horizontal wind speed near the DTU turbine hub height of 119 m, for the entire 10-minute simulation on a 1 km x 1 km grid. The four dashed circles represent the reference radii of 10, 12, 15 and 20 km.**

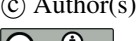



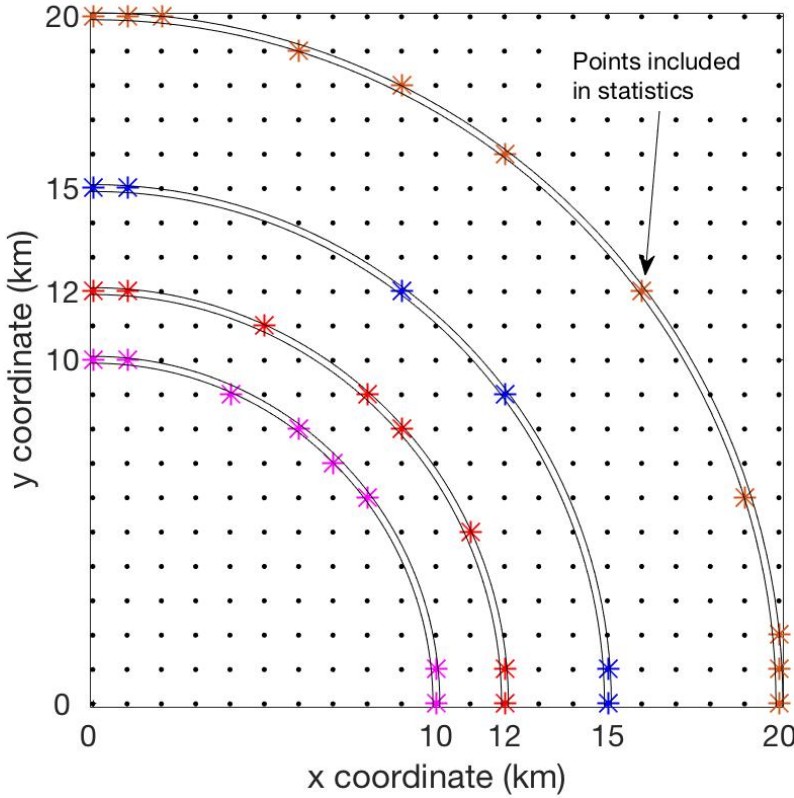


**Figure 2: Virtual towers (black circles) in one quadrant of the LES domain horizontally spaced at 1 km resolution. Selected virtual towers are included in the analysis when they are within 100 m of the reference radii: 10 km (magenta), 12 km (red), 15 km (blue) and 20 km (orange). Reference radii are measured from the hurricane center.**

**2.2 Wind speed gusts**

The 3-sec gust is defined in the design requirements by the (IEC, 2009) as the peak 3-sec average wind speed within a 10-min interval. Associated with the gust is the gust factor, defined as the ratio of the maximum 3-sec gust to the local mean wind speed (IEC, 2009). The maximum gust considered by the IEC design requirements for a Class I turbine is 70 m/s with a 50 m/s mean wind speed at the hub height, corresponding to a gust factor of 1.4. During Typhoon Maemi in Japan, peak gusts of 74 m/s were measured when the mean wind speed was only 38 m/s,

corresponding to a gust factor of 1.95, much higher than that considered by IEC (Ishihara et al., 2005). Typhoon Maemi caused significant damage to all the wind turbines in an onshore coastal wind farm, underscoring the importance of revisiting gust factor specifications. Similarly, two onshore wind farms in Puerto Rico (Santa Isabel and Punta Lima) were affected by hurricane Maria in 2017, a Category 4 hurricane at landfall. Santa Isabel missed the eyewall of the hurricane and survived relatively intact while Punta Lima, which had a direct eyewall hit,

experienced significant destructive damage (Gallucci, 2018; Kelley, 2017; Rocky Mountain Institute, 2017).




Gusts in the LES simulations exceed those considered by the IEC, which is expected since the simulated storm is Category 5 with mean wind speeds larger than those considered in the IEC design requirements. The maximum gust factor in the simulated hurricane, outside of the quiescent zone within the eye, is approximately 1.7, substantially in excess of the IEC recommended gust factor or 1.4. These results, along with more detailed discussion of the wind

speed and gust features of the simulated storm, are available in (Worsnop et al., 2017a).

### 2.3 Change in wind direction and veer

Wind direction is also an important consideration in evaluating the effects of hurricanes on offshore wind turbines. Large wind direction changes can be particularly critical for wind turbines during extreme events, because a loss of connection to the grid prevents wind turbines from being able to yaw into the wind direction or because the wind

direction changes too rapidly for even a functioning yaw control system to accommodate.  For example, simulations performed by the authors indicate a yaw rate of 1.0-1.3 deg s$^{-1}$ for the NREL 5MW reference turbine. The design requirements in IEC 61400-3 (IEC, 2009) require that a turbine in the parked condition during a storm consider two loading conditions relevant to wind direction changes. In the first (Design Load Case 6.1), loading from a misalignment between the wind direction and the rotor plane of ± 15°, if using the steady extreme wind model, or of

± 8°, if using the turbulent extreme wind model, must be considered. In the second (Design Load Case 6.2), loading from misalignments of ± 180° must be considered: this situation corresponds to the situation where a turbine has lost control of the rotor orientation due to power loss. The first condition is considered a normal event and has a load factor of 1.35 (the factor by which loads are increased to help ensure safety), while the second condition is considered an abnormal event and has a load factor of 1.10. A practical way to contextualize the results of this study is to compare

the wind direction changes in the LES wind field to the magnitudes of misalignment considered by IEC during conditions when rotor control is maintained and when it is lost.

(Worsnop et al., 2017a) provide distributions of the 10-s and 30-s maximum direction change at hub height at a range of radii from the storm center.  At radii outside the quiescent zone, mean maximum direction change ranges from approximately 15° to 20° and maximum direction changes approach 35°. To illustrate the character of the wind

direction in time, Fig. 3 shows 10-minute time series of the maximum direction change over periods of 10 s and 30 s at radii of 10 km, 12 km, 15 km and 20 km. The direction changes exhibit very rapid fluctuations, and the maximum values approach 30° for both the 10 s and 30 s periods at 10 km and 20 km radii. Rapid wind direction changes of that magnitude at such high wind speeds are likely to be difficult for even a functioning yaw control system to manage.





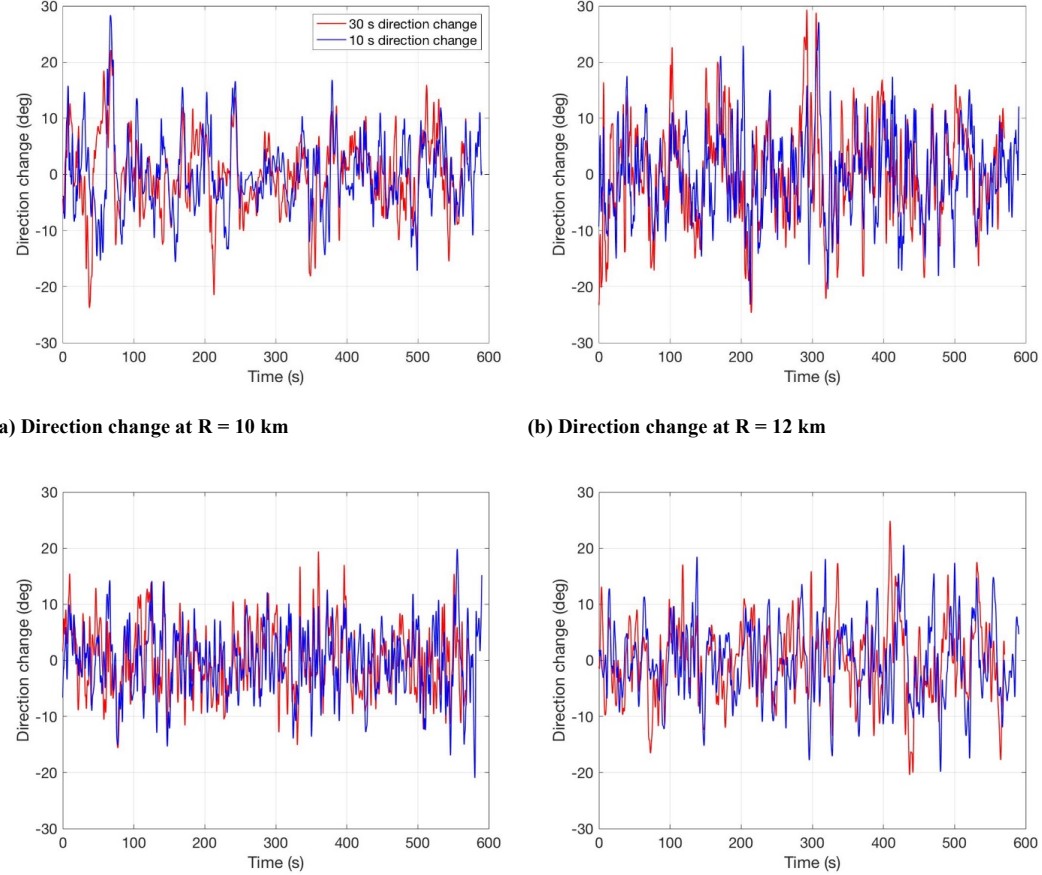

**(a) Direction change at R = 10 km**

**(b) Direction change at R = 12 km**

**(c) Direction change at R = 15 km**

**(d) Direction change at R = 20 km**

**Figure 3: Time series of maximum wind direction change at various radii (R) of the hurricane at hub height: (a) inner edge of eyewall R = 10 km, (b) inside eyewall R = 12 km, (c) outer edge of eyewall R = 15 km and (d) outside of the eyewall R = 20 km. The red and the blue lines represents the maximum wind direction change over moving intervals of 30 s (red) and 10 s (blue).**

In addition to direction change over time at hub height, wind direction can also vary across a vertical profile at the
same time instant. This vertical variation in direction, called veer, is next characterized for the simulated hurricane wind fields. Although numerous onshore observations of veer indicate its prevalence, especially at night, and veer associated with nocturnal low-level jets has been implicated in damage to onshore wind turbines (Kelley et al., 2006), as well as in affecting turbine power production (Vanderwende and Lundquist, 2012), veer is not currently considered in wind turbine design specifications, either onshore or offshore. Veer may cause additional demand on the blades
since the rotor is only yawed and feathered relative to a single wind direction, usually sensed at the nacelle (Giebel and Gryning, 2004).



To characterize the veer quantitatively, four reference veer profiles are defined using the wind direction at the LES grid points closest to the hub height, rotor top and bottom elevations of the DTU reference wind turbine (117.19 m, 210.94 m, 39.06 m). The profiles (INC, DEC, VEE, INV) are defined according to the wind directions at the top of the rotor disk $\theta_{top}$, the hub height $\theta_{hub}$ and the bottom of the rotor disk $\theta_{bot}$, as shown in Table 1, and every instant of the simulation time histories is categorized into one of these profiles. Table 1 also shows the frequency of occurrence of each of the veer shapes and shows that INC, one of the monotonically varying veer profiles, dominates. Different veer profiles may induce different load conditions on the blades, rotor, and tower.

**Table 1: Percentage occurrence of the four different veer profiles at each reference radius.**

|  | R = 10 km | R = 12 km | R = 15 km | R = 20 km |
|---|---|---|---|---|
| INC (monotonic increase) = $\theta_{top} < \theta_{hub} < \theta_{bot}$ | 44 % | 57 % | 64 % | 67 % |
| DEC (monotonic decrease) = $\theta_{top} > \theta_{hub} > \theta_{bot}$ | 9 % | 4 % | 4 % | 3 % |
| VEE (V) = $\theta_{bot}$ , $\theta_{top} > \theta_{hub}$ | 25 % | 17 % | 13 % | 14 % |
| INV (inverse V) = $\theta_{bot}$ , $\theta_{top} < \theta_{hub}$ | 22 % | 22 % | 19 % | 16 % |

Although the wind directions at the rotor top, hub height and rotor bottom define the overall shape of the veer profile, the LES provide the wind direction at a series of vertical points spaced at 15.625 m. The actual change in wind direction vertically across the rotor disk may be nonlinear (Fig. 4) as seen by the profiles with maximum veer within each profile type at each radius. The veer magnitude is defined as

$$\theta_{veer} = \left|\theta_{top} - \theta_{hub}\right| + \left|\theta_{hub} - \theta_{bot}\right|.$$

Direction changes of over 30° between the top and bottom of the rotor can occur within the hurricane boundary layer, particularly in the eyewall region.

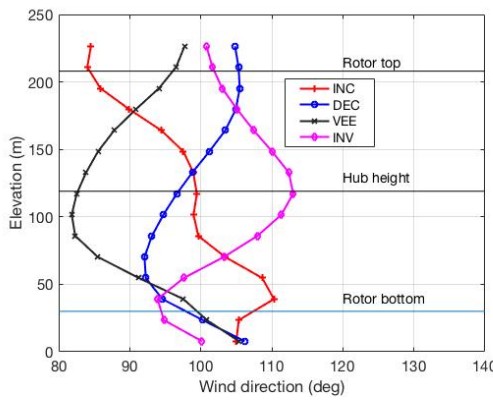
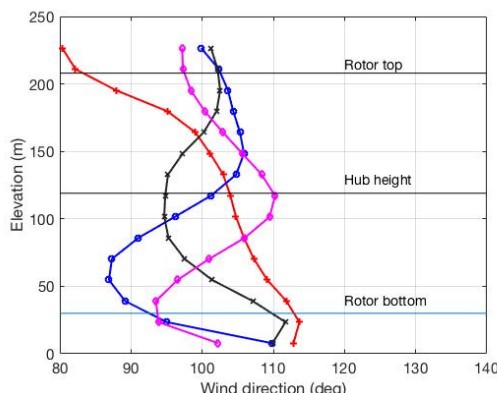

(a) Max veer profiles at R = 10 km          (b) Max veer profiles at R = 12 km

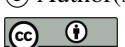



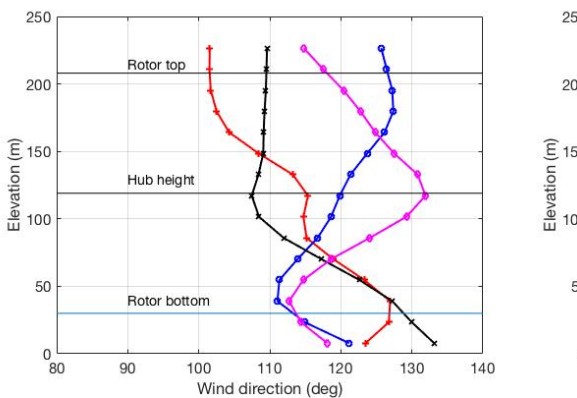
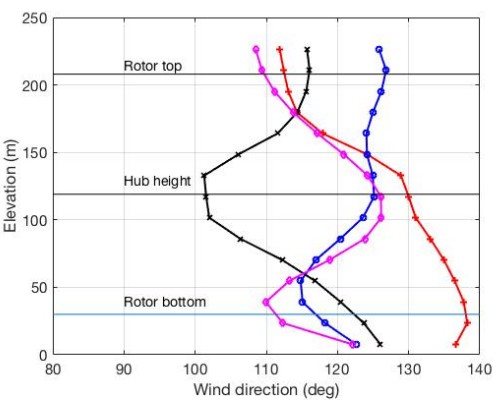

**(c) Max veer profiles at R = 15 km**     **(d) Max veer profiles at R = 20 km**

**Figure 4: Maximum instantaneous veer profile for the four profile shapes at four radii from the hurricane center: (a) internal eyewall boundary R = 10 km, (b) inside eyewall R = 12 km, (c) external eyewall boundary R = 15 km and (d) outside of the eyewall R = 20 km. The hub height and rotor top and bottom of the theoretical DTU reference turbine, at elevations of 119m, 208 m and 30 m, respectively are indicated with horizontal lines.**

## 3 Generating hurricane wind fields in TurbSim

### 3.1 TurbSim inputs

To simulate a wind turbine in FAST subjected to hurricane wind conditions, TurbSimv2 is used to generate wind input files that are compatible with FAST and which capture the key characteristics of the wind fields produced by the LES simulations. TurbSim is a full-field, turbulent-wind simulator developed by the National Renewable Energy Laboratory (Jonkman and Buhl Jr., 2005). Using statistics from the LES data at each of the four radii defined in Section 2, wind input files representing the wind field at each radius were generated. Each wind input file is one-hour long, consistent with IEC standards for wind turbine structural design and load calculations (IEC, 2009).

TurbSim requires inputs of mean reference wind speed and turbulence intensity at the turbine's hub height, spectrum (Kaimal in this case), wind shear profile, wind veer profile, and coherence exponent (applied to eq. 18 in (Jonkman and Buhl Jr., 2009). Given these parameters, TurbSim models turbulence as a Gaussian process with no skewness and kurtosis equal to 3, while the turbulence in the LES wind field can be non-Gaussian.

### 3.2 TurbSim and LES data comparisons

The mean reference wind speed and turbulence intensity were determined using the LES data at a height nearest the hub height at $z$ = 117.19 m. Skewness and kurtosis are not TurbSim input parameters, and therefore comparison between the values of skewness and kurtosis indicate the goodness of fit of the marginal distribution of the TurbSim wind fields relative to those in the LES simulations.

While the mean, turbulence intensity, and kurtosis of the two wind fields match almost exactly, the largest differences appear in the skewness (Table 2). (Mean and turbulence intensity should match since these two values are prescribed

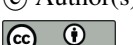



to TurbSim.) TurbSim assumes normally distributed turbulence statistics, and thus the skewness is nearly zero in all
cases. The LES wind field has small but non-negligible skewness, resulting in large percentage differences between
the two sets of wind files, though the absolute differences are small.

**Table 2: Comparison of LES and TurbSim wind field statistics.**

|  |  | Mean, $U$ (m/s) | STD (m/s) | TI (%) | Skewness, $S$ (-) | Kurtosis, $k$ (-) |
|---|---|---|---|---|---|---|
| R = 10 km | LES | 72.2 | 6.6 | 9.2 | 0.80 | 3.6 |
|  | TurbSim | 72.2 | 6.6 | 9.1 | 0.02 | 2.8 |
|  | % error | -0.01 | -1.2 | -1.3 | -97 | -23 |
| R = 12 km | LES | 88.4 | 6.5 | 7.3 | -0.2 | 2.9 |
|  | TurbSim | 88.4 | 6.4 | 7.2 | 0.02 | 2.8 |
|  | % error | 0.06 | -1.4 | -1.4 | -93 | -2.8 |
| R = 15 km | LES | 81.2 | 6.4 | 7.9 | -0.2 | 2.8 |
|  | TurbSim | 81.3 | 6.4 | 7.8 | 0.02 | 2.8 |
|  | % error | 0.10 | -1.3 | -1.3 | -89 | -0.12 |
| R = 20 km | LES | 65.7 | 5.4 | 8.2 | 0.40 | 3.0 |
|  | TurbSim | 65.8 | 5.3 | 8.1 | 0.03 | 2.8 |
|  | % error | 0.06 | -1.3 | -1.2 | -94 | -8.8 |

The wind shear profiles from LES and TurbSim are indistinguishable (Fig. 5). The wind shear profile for each radii
was specified using the mean horizontal wind speed from LES for all grid cells between the top and bottom of the
rotor disk. These shear profiles were then specified in TurbSim's "User Defined Profile," available in TurbSimv2. In
both cases, the profile at the 10-km radius deviates from a standard power law profile.

Two cases were considered for the wind veer profile: a baseline case with no veer and a case with veer. The veer
profiles are specified in the TurbSim input file. In the "no-veer" case, the wind direction is 0 degrees at all heights. In
the veer case, at each radii, the 4 veer profiles shown in Fig. 4 are specified. As in the case of wind shear, the LES and
TurbSim veer profiles match exactly, and thus are not shown here.





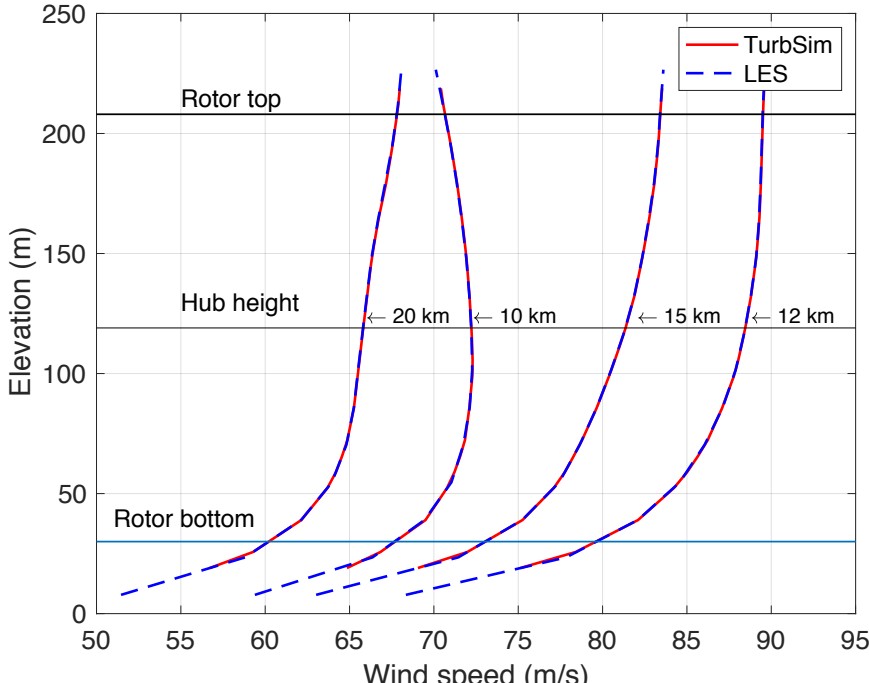

**Figure 5: Wind shear profile: the mean wind shear profile at R = 10, 12, 15 and 20 km for both the LES and TurbSim wind field simulations. For reference, the three horizontal lines indicate the top and bottom of the rotor disk, at 208 m and 30 m, respectively and the hub height at 119 m.**

The power spectral density compares well between LES and TurbSim (Fig. 6). The hub-height wind speed power spectral density created by TurbSim is modeled with a Kaimal spectrum. The temporal resolution of the LES data and the implicit subgrid-scale filter causes the power to fall off at high frequencies, more quickly than the TurbSim data. Aside from this expected difference, the plots show a reasonable agreement between the two datasets at all four radii.





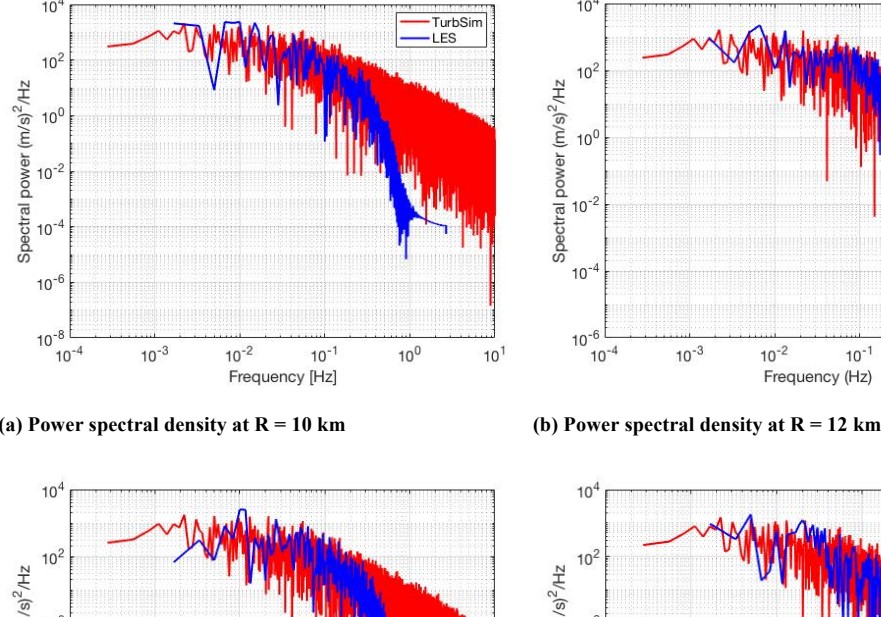

(a) Power spectral density at R = 10 km        (b) Power spectral density at R = 12 km

(c) Power spectral density at R = 15 km        (d) Power spectral density at R = 20 km

**Figure 6: Power spectral density comparisons of the LES and TurbSim 10-minute wind-field simulations at hub height, at four different radii from the hurricane center.**


The coherence exponent controls the degree of spatial coherence between neighboring points as a function of distance between points (Jonkman and Buhl Jr., 2009). Coherence in the hurricane boundary layer may persist at larger horizontal separations than seen in non-hurricane boundary layers (Worsnop et al., 2017b). In TurbSim, the vertical coherence of the wind speed is defined for the x, y, and z

components of wind individually by

$$Coh_{i,j} = \exp\left(-a\left(\frac{r}{z_m}\right)^{CohExp}\sqrt{\left(\frac{fr}{\overline{u}_m}\right)^2 + (br)^2}\right) \tag{1}$$




where $r$ is the vertical distance between points $i$ and $j$, $f$ is the cyclic frequency, $CohExp$ is the coherence exponent input parameter, $z_m$ and $\bar{u}_m$ are the mean height and wind speed of points $i$ and $j$, while $a$ and $b$ are the spatial coherence decrement and offset parameter for the component of the wind speed under consideration.

The appropriate value for the coherence exponent was evaluated by calculating the correlation coefficient between the hub height wind speed time series and the wind speed time series at all other heights in the LES data. Recall that the vertical spacing between grid points in the LES data is 15.6 m. The option for setting the coherence exponent value was limited by the TurbSim software. Through a process of trial and error, a single coherence exponent of 0.85 was selected for all locations for consistency between simulations, which is near the maximum allowable value of 1.0.

This selection represents a significant increase in coherence compared to the default value of 0.0, but previous work indicates that the hurricane boundary layer likely contains coherent structures such as roll vortices that would increase the coherence of the flow (Worsnop et al., 2017b).

As the coherence exponent increases, the correlation between the wind speed time series at spatially separated points also increases, increasing the variance of the total blade load and, therefore, the maximum load. The LES data tends

to have higher spatial correlation in the vertical direction than the TurbSim data (Fig. 7). However, the default TurbSim coherence exponent of 0.0 causes poorer agreement as the vertical correlation within the TurbSim data drops off even more rapidly with distance than is seen in the LES data. Capturing the spatial coherence of the wind field to the greatest degree possible is important for estimating structural response because wind fields with longer range coherence would be expected to generate greater variability in structural loads. This difference occurs because the blades act as

lengthwise integrators of the local aerodynamic forces.

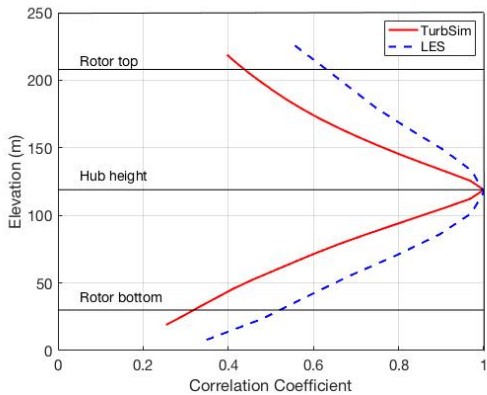
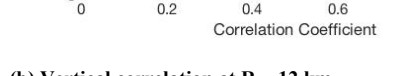

(a) Vertical correlation at R = 10 km          (b) Vertical correlation at R = 12 km




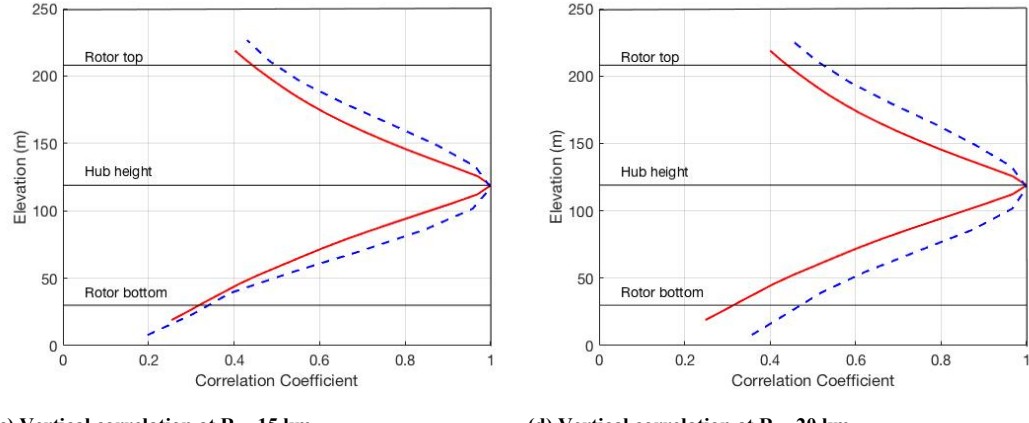

**(c) Vertical correlation at R = 15 km**  **(d) Vertical correlation at R = 20 km**

**Figure 7: Profiles of correlation coefficient between the 10-min horizontal wind speed time series at each elevation and the 10-min horizontal wind speed time series at hub height for TurbSim (red) and LES (blue dashed) simulations, at four different radii from the hurricane center.**

## 4 Structural effects of hurricane winds on wind turbines

### 4.1 FAST Simulations

Four cases were considered for each of the four radii: baseline hurricane winds (BASE), hurricane winds with veer (VEER), misaligned hurricane winds (MISAL), and, for comparison, winds at rated power (RATED). The BASE case represents an idling turbine with a wind field corresponding to the mean wind speed and turbulence intensity from the LES simulation but without any veer or direction change. The VEER case considers the largest magnitude veer case as described in Section 3 with the veer profile applied at 1-m vertical increments. The MISAL case considers the baseline wind with a yawed turbine to evaluate the impact of rapid direction change. The degree of yaw misalignment is determined for each location based on the maximum 10-sec direction change described in Section 2. Yaw misalignment may occur if the wind changes direction more quickly than the yaw controller can adjust the nacelle. For these three cases (BASE, VEER, and MISAL), the turbine blades are pitched to feather, the rotor is set to idle, and the generator is turned off. For the RATED case, an operational turbine is simulated. The rated wind speed case considers the turbine operating in conditions with a mean wind speed of 11.4 m/s, a turbulence intensity of 10%, power law exponent of 0.1, and a coherence exponent of 0.0. The blade pitch and generator controllers are active during simulations. All simulations are one hour long to be consistent with IEC standards for wind turbine structural design and load calculations (IEC 2009. Table 3 summarizes the wind field characteristics for the full set of simulation cases.

**Table 3: Wind field characteristics for suite of simulation cases.**





| Condition (short form label) | Reference radius R (km) | Mean wind speed (m/s) (Turbulence Intensity) | Wind shear profile or power law exponent | Coherence exponent | Veer (deg) | Yaw misalignment (deg) | Operating state |
|---|---|---|---|---|---|---|---|
| Baseline hurricane (BASE) | 10 | 72.2 (9.27%) | See Fig. 5 | 0.85 | None | None | Idling |
| | 12 | 88.4 (7.38%) | | | | | |
| | 15 | 81.3 (7.97%) | | | | | |
| | 20 | 65.8 (8.25%) | | | | | |
| Hurricane w/ veer (VEER) | 10 | 72.2 (9.27%) | See Fig. 5 | 0.85 | INC profile from Fig. 4 | None | Idling |
| | 12 | 88.4 (7.38%) | | | | | |
| | 15 | 81.3 (7.97%) | | | | | |
| | 20 | 65.8 (8.25%) | | | | | |
| Misaligned hurricane (MISAL) | 10 | 72.2 (9.27%) | See Fig. 5 | 0.85 | None | 28.4 | Idling |
| | 12 | 88.4 (7.38%) | | | | 27.1 | |
| | 15 | 81.3 (7.97%) | | | | 19.9 | |
| | 20 | 65.8 (8.25%) | | | | 20.6 | |
| Rated (RATED) | N/A | 11.4 (0.10%) | 0.10 | 0.00 | None | None | Operating |

## 4.2 Maximum turbine structural loads

When the BASE, VEER, or MISAL cases are applied to the turbine, the tower demands in the side-to-side (SS)
direction and the blade demands in the rotor plane increase compared to the RATED case. The tower demands in the
fore-aft (FA) direction and the blade demands out of the rotor plane are either reduced or do not increase significantly
because the non-rated wind cases have the blades pitched to feather, reducing the impact of the wind in those directions.
The MISAL wind causes the largest increase in loads, followed by VEER, then BASE. The increased loads due to
wind misalignment and veer emphasize the importance of considering misalignment and veer when evaluating turbine
structural response (Tables 4 through 8).

Hurricane wind field characteristics such as VEER and MISAL cause substantial increases in certain turbine (blade
and tower) loads when compared to BASE and RATED cases (Tables 4 through 8). The BASE loads are presented as




scale factors relative to the RATED loads in Table 4. The VEER loads appear as scale factors of RATED in Table 5
and scaled to BASE in Table 6. The MISAL loads appear as scale factors of RATED in Table 7 and scaled to BASE

in Table 8.

**Table 4: Maxima of blade and tower structural response to BASE hurricane wind case. Scale factor (S.F.) indicates magnitude of response relative to RATED. Numerical values of scale factors are not provided when not appropriate (for blade deflections the blade is oriented differently in RATED and scaling of FA/SS tower base moment ratios is not a meaningful measure of response).**

|  |  | RATED | BASE |  |  |  |  |  |  |  |
|---|---|---|---|---|---|---|---|---|---|---|
|  |  | 11.4 m/s | R = 10 km |  | R = 12 km |  | R = 15 km |  | R = 20 km |  |
|  |  | Max | Max | S.F. | Max | S.F. | Max | S.F. | Max | S.F. |
| Out of Plane Blade Tip Deflection | [m] | 11 | 0.8 | - | 1.6 | - | 1.3 | - | 0.5 | - |
| In Plane Blade Tip Deflection | [m] | 1.7 | 9.0 | - | 11 | - | 10 | - | 7.3 | - |
| FA Tower Base Moment | [kNm] | 2.1E+05 | 3.7E+05 | 1.7 | 4.8E+05 | 2.3 | 4.2E+05 | 2.0 | 2.8E+05 | 1.3 |
| SS Tower Base Moment | [kNm] | 2.0E+04 | 1.4E+05 | 6.8 | 1.5E+05 | 7.4 | 1.5E+05 | 7.2 | 1.1E+05 | 5.4 |
| FA/SS Tower Base Moment Ratio | [-] | 10 | 2.7 | - | 3.2 | - | 2.9 | - | 2.5 | - |
| Tower Base Resultant Moment | [kNm] | 2.1E+05 | 3.7E+05 | 1.7 | 4.8E+05 | 2.2 | 4.2E+05 | 2.0 | 2.8E+05 | 1.3 |


For selected loads, the BASE loads surpass those of the RATED case. While the BASE out-of-plane blade loads in Table 4 are less than the RATED loads, the in-plane values exceed what is expected of rated operation. Similarly, the side-to-side tower BASE moment increase is approximately three times greater than the fore-aft moment increase. However, the side-to-side moment for the RATED case is a factor of ten lower than the fore-aft moment, so the
resultant moment for the BASE cases is around two times the moment for RATED. The simulated storm is a Category 5 and therefore increased loads relative to RATED should occur.

**Table 5 Maxima of blade and tower structural response to VEER case. Scale factor (S.F) indicates magnitude of response relative to RATED. Cells highlighted in gray indicate values that exceed Eurocode strength basis value for the tower (5.4 x 10^5 kNm) or the 18 m deflection limit on the blade tip. Numerical values of scale factors are not provided when not**
**appropriate (for blade deflections the blade is oriented differently in the rated case and scaling of FA/SS tower base moment ratios is not a meaningful measure of response).**

|  |  | RATED | VEER |  |  |  |  |  |  |  |
|---|---|---|---|---|---|---|---|---|---|---|
|  |  | 11.4 m/s | R = 10 km |  | R = 12 km |  | R = 15 km |  | R = 20 km |  |
|  |  | Max | Max | S.F. | Max | S.F. | Max | S.F. | Max | S.F. |
| Out of Plane Blade Tip Deflection | [m] | 10.8 | 2.4 | - | 3.5 | - | 3.9 | - | 3.6 | - |
| In Plane Blade Tip Deflection | [m] | 1.7 | 25 | - | 34 | - | 35 | - | 24 | - |
| FA Tower Base Moment | [kNm] | 2.1E+05 | 3.3E+05 | 1.6 | 4.4E+05 | 2.1 | 3.7E+05 | 1.7 | 2.3E+05 | 1.1 |
| SS Tower Base Moment | [kNm] | 2.0E+04 | 3.3E+05 | 16 | 4.0E+05 | 20 | 5.1E+05 | 25 | 3.8E+05 | 19 |
| FA/SS Tower Base Moment Ratio | [-] | 10 | 1.0 | - | 1.1 | - | 0.7 | - | 0.6 | - |
| Tower Base Resultant Moment | [kNm] | 2.1E+05 | 3.5E+05 | 1.6 | 4.7E+05 | 2.2 | 5.3E+05 | 2.5 | 3.8E+05 | 1.8 |

**Table 6 Maxima of blade and tower structural response to VEER case. Scale factor (S.F) indicates magnitude of response relative to BASE. Cells highlighted in gray indicate values that exceed Eurocode strength basis value for the tower (5.4 x 10^5 kNm) or the 18 m deflection limit on the blade tip. Numerical values of scale factors are not provided when not appropriate (for blade deflections the blade is oriented differently in the rated case and scaling of FA/SS tower base moment ratios is not a meaningful measure of response).**

|  |  | R = 10 km |  |  | R = 12 km |  |  | R = 15 km |  |  | R = 20 km |  |  |
|---|---|---|---|---|---|---|---|---|---|---|---|---|---|
|  |  | BASE | VEER |  | BASE | VEER |  | BASE | VEER |  | BASE | VEER |  |
|  |  | Max | Max | S.F. | Max | Max | S.F. | Max | Max | S.F. | Max | Max | S.F. |
| Out of Plane Blade Tip Deflection | [m] | 0.8 | 2.4 | 3.0 | 1.6 | 3.5 | 2.2 | 1.3 | 3.9 | 3.0 | 0.5 | 3.6 | 6.9 |
| In Plane Blade Tip Deflection | [m] | 9.0 | 25 | 2.7 | 11 | 34 | 3.2 | 10 | 35 | 3.5 | 7.3 | 24 | 3.3 |
| FA Tower Base Moment | [kNm] | 3.7E+05 | 3.3E+05 | 0.9 | 4.8E+05 | 4.4E+05 | 0.9 | 4.2E+05 | 3.7E+05 | 0.9 | 2.8E+05 | 2.3E+05 | 0.8 |
| SS Tower Base Moment | [kNm] | 1.4E+05 | 3.3E+05 | 2.4 | 1.5E+05 | 4.0E+05 | 2.7 | 1.5E+05 | 5.1E+05 | 3.5 | 1.1E+05 | 3.8E+05 | 3.5 |
| FA/SS Tower Base Moment Ratio | [-] | 2.7 | 1.0 | - | 3.2 | 1.1 | - | 2.9 | 0.7 | - | 2.5 | 0.6 | - |
| Tower Base Resultant Moment | [kNm] | 3.7E+05 | 3.5E+05 | 0.9 | 4.8E+05 | 4.7E+05 | 1.0 | 4.2E+05 | 5.3E+05 | 1.3 | 2.8E+05 | 3.8E+05 | 1.4 |





When including veer in the wind speed profile veer, loads increase. The loads for the VEER case in Tables 5 and 6 follow a similar trend as the BASE case; however, the veer causes the load increases to be substantially higher, especially in the side-to-side and in-plane directions. The introduction of veer causes all side-to-side, in-plane, and resultant loads to exceed the RATED loads. When compared to the BASE case, the resultant tower base moment in

VEER is nearly the same or slightly greater. The resultant blade root bending moment for VEER was on average 3.5 times greater than even the BASE case and 1.75 greater than RATED. Veer has a significant effect on blade deflections and blade root moments since a blade can be pitched to only a single angle, and that angle cannot correspond to a feathered state when wind direction changes along the length of the blade.

**Table 7 Maxima of blade and tower structural response to MISAL case. Scale factor (S.F) indicates magnitude of response relative to RATED. Cells highlighted in gray indicate values that exceed Eurocode strength basis value for the tower (5.4 x 10$^5$ kNm) or the 18 m deflection limit on the blade tip. Numerical values of scale factors are not provided when not appropriate (for blade deflections the blade is oriented differently in the rated case and scaling of FA/SS tower base moment ratios is not a meaningful measure of response).**

| | | RATED | MISAL | | | | | | | | |
| | | 11.4 m/s | R = 10 km | | R = 12 km | | R = 15 km | | R = 20 km | |
| | | Max | Max | S.F. | Max | S.F. | Max | S.F. | Max | S.F. |
|---|---|---|---|---|---|---|---|---|---|---|
| Out of Plane Blade Tip Deflection | [m] | 11 | 7.1 | - | 17 | - | 8.5 | - | 5.3 | - |
| In Plane Blade Tip Deflection | [m] | 1.7 | 35 | - | 50 | - | 42 | - | 26 | - |
| FA Tower Base Moment | [kNm] | 2.1E+05 | 8.0E+05 | 3.8 | 5.9E+05 | 2.8 | 4.2E+05 | 2.0 | 2.7E+05 | 1.3 |
| SS Tower Base Moment | [kNm] | 2.0E+04 | 1.8E+06 | 88 | 1.2E+06 | 57 | 1.1E+06 | 54 | 7.2E+05 | 35 |
| FA/SS Tower Base Moment Ratio | [-] | 10.5 | 0.4 | - | 0.5 | - | 0.4 | - | 0.4 | - |
| Tower Base Resultant Moment | [kNm] | 2.1E+05 | 1.8E+06 | 8.4 | 1.2E+06 | 5.5 | 1.1E+06 | 5.3 | 7.4E+05 | 3.5 |

**Table 8 Maxima of blade and tower structural response to MISAL case. Scale factor (S.F) indicates magnitude of response relative to BASE. Cells highlighted in gray indicate values that exceed Eurocode strength basis value for the tower (5.4 x 10$^5$ kNm) or the 18 m deflection limit on the blade tip. Numerical values of scale factors are not provided when not appropriate (for blade deflections the blade is oriented differently in the rated case and scaling of FA/SS tower base moment ratios is not a meaningful measure of response).**

| | | R = 10 km | | | R = 12 km | | | R = 15 km | | | R = 20 km | | |
| | | BASE | MISAL | | BASE | MISAL | | BASE | MISAL | | BASE | MISAL | |
| | | Max | Max | S.F. | Max | Max | S.F. | Max | Max | S.F. | Max | Max | S.F. |
|---|---|---|---|---|---|---|---|---|---|---|---|---|---|
| Out of Plane Blade Tip Deflection | [m] | 0.8 | 7.1 | 8.6 | 1.6 | 17 | 11 | 1.3 | 8 | 6.6 | 0.5 | 5.3 | 10 |
| In Plane Blade Tip Deflection | [m] | 9.0 | 35 | 3.8 | 11 | 50 | 4.7 | 10 | 42 | 4.2 | 7.3 | 26 | 3.6 |
| FA Tower Base Moment | [kNm] | 3.7E+05 | 8.0E+05 | 2.2 | 4.8E+05 | 5.9E+05 | 1.2 | 4.2E+05 | 4.2E+05 | 1.0 | 2.8E+05 | 2.7E+05 | 1.0 |
| SS Tower Base Moment | [kNm] | 1.4E+05 | 1.8E+06 | 13 | 1.5E+05 | 1.2E+06 | 7.7 | 1.5E+05 | 1.1E+06 | 7.5 | 1.1E+05 | 7.2E+05 | 6.6 |
| FA/SS Tower Base Moment Ratio | [-] | 2.7 | 0.4 | - | 3.2 | 0.5 | - | 2.9 | 0.4 | - | 2.5 | 0.4 | - |
| Tower Base Resultant Moment | [kNm] | 3.7E+05 | 1.8E+06 | 4.8 | 4.8E+05 | 1.2E+06 | 2.5 | 4.2E+05 | 1.1E+06 | 2.7 | 2.8E+05 | 7.4E+05 | 2.6 |


The misaligned wind causes even greater load increases than the VEER case and does so for both the blades and the tower. Only the out-of-plane tip deflection is still lower than that of the RATED case. The in-plane tip deflection is around 20 times greater than the RATED in-plane tip deflection, reaching 50 m at radial position of 12 km. The side-to-side tower base moment also show substantially higher results with a moment 90 times greater than rated at a radial

position of 10 km. When presented as a scale factor applied to the BASE case, the impact of potential yaw misalignment is particularly apparent with an increase on average for every load assessed. The resultant blade root bending moment is approximately six times that of the BASE case and five times greater than the RATED case, while the resultant tower base moment is approximately 2.5 times greater than the BASE case and 5.5 times greater than the RATED case.





**4.3 Timeseries of select turbine structural loads at R = 12 km**

Time series provide an indication of how the load histories vary under the different wind cases with more detail than the maximum responses given in the preceding tables (Figs. 8-10). The 12-km radius, within the eyewall, has the highest mean wind speed and shows the greatest variation in loads as the wind changes. The misaligned flow (MISAL) causes the largest standard deviation and mean for in-plane tip deflection (purple line in Fig. 8), blade root resultant

moment (purple line in Fig. 9), and tower base resultant moment (purple line in Fig. 10). The VEER case causes a greater mean than RATED for all three loads presented but a similar standard deviation for the tower base resultant moment. As presented above, the BASE wind (blue lines in Fig. 8-10) causes an increase in in-plane tip deflection and tower base resultant moment compared to RATED, but the mean and maximum blade root resultant moment decrease for BASE, while the standard deviation increases.

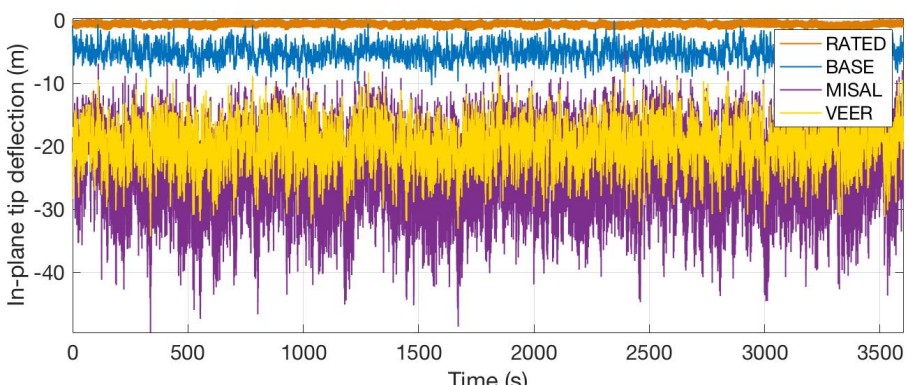


**Figure 8: Time series of in-plane blade tip deflection for turbine simulations at a radial position of 12 km with rated wind (RATED), baseline wind (BASE), wind with veer (VEER), and misaligned wind (MISAL).**

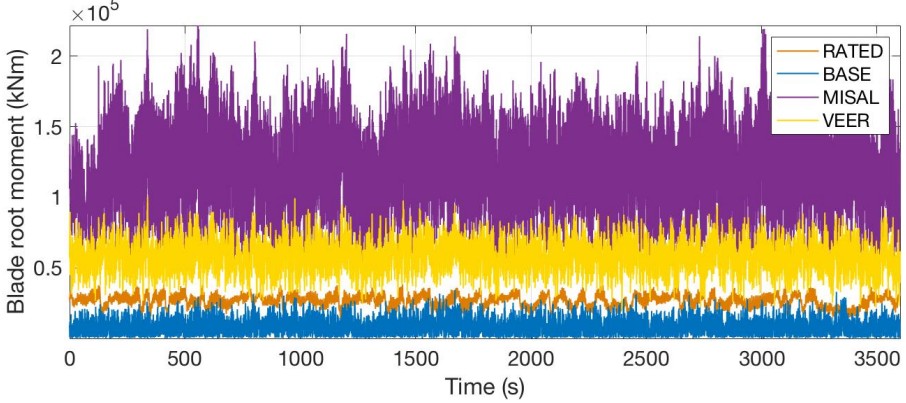

**Figure 9: Time series of the resultant moment at the blade root for turbine simulations at a radial position of 12 km with**
**rated wind (RATED), baseline wind (BASE), wind with veer (VEER), and misaligned wind (MISAL).**

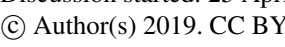


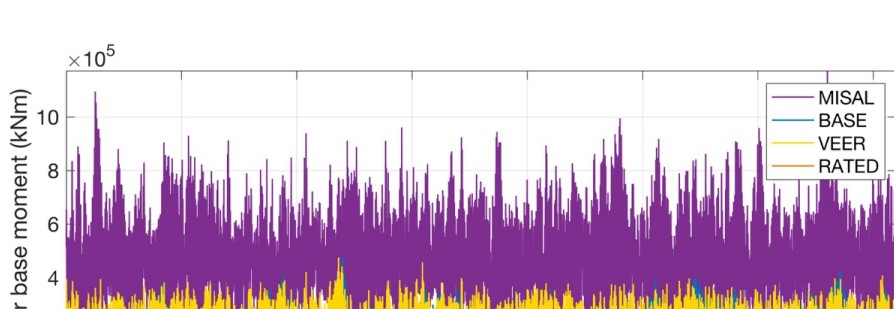

**Figure 10: Time series of tower base resultant moment for turbine simulations at a radial position of 12 km with rated wind (RATED), baseline wind (BASE), wind with veer (VEER), and misaligned wind (MISAL).**

**5 Conclusions**

With the expansion of offshore wind off the US East Coast, critical questions emerge regarding hurricane-induced loads on offshore wind turbines. Given the paucity of high-rate observations of winds and turbulence offshore in turbine rotor altitudes, we have integrated atmospheric large-eddy simulations (LES) of an idealized Category 5 hurricane with the CM1 model into the engineering wind field simulator TURBSIM to estimate loads on a 10MW turbine using FAST. In particular, we evaluate how turbine and tower structures respond to flows characteristic of the

hurricane boundary layer, particularly the eyewall, such as high wind speeds, veer in wind profiles, and rapid wind direction changes.

TURBSIM wind fields can represent the mean wind speed, turbulence intensity, power spectra, veer, coherence, and direction change (represented as yaw misalignment) as they were calculated in CM1's LES of a Category 5 hurricane. Significant veer across the rotor disk (approximately 30$^\text{o}$) and rapid direction changes at hub height (up to 30$^\text{o}$ in 10 s)

occur in the LES simulations. Further, the hurricane boundary layer appears to impose a much stronger spatial coherence than is normally assumed, perhaps due to coherent structures in the eyewall. Simulations show that veer and direction change can dramatically increase loads on the blades and tower, in some cases by factors of five or more. In general, veer primarily increases blade loads while rapid direction changes at hub height amplify both blade and tower loads. Hurricane loads on wind turbines may therefore exceed those loads predicted when hurricane wind fields

are simulated without the inclusion of veer and rapid direction change. Yaw misalignment cases are specified in design standards such as IEC 61400-3, but for normal operating cases the magnitude of the misalignment is only 8$^\text{o}$ or 15$^\text{o}$ depending on the corresponding wind model. The results of this paper, however, show that much larger misalignments may occur due to rapid wind direction changes

The loads calculated here should motivate further investigation into how the hurricane boundary layer affects wind

turbines. This Category 5 scenario represents a worst-case scenario. Simulations of weaker hurricanes, which would





be more frequently expected, must assess more likely risks to large-scale deployments of wind turbines. Also needed are assessments of the likelihood of these conditions in order to dictate design conditions as part of a probabilistic design basis.

Refinements to the LES would also provide insight. The LES simulations are conducted assuming that the hurricane is over open ocean. The onshore version of the DTU 10-MW turbine is used along with these offshore wind fields to isolate the effect of hurricane wind field characteristics in absence of other structural loading from waves. These LES are not coupled with a wave model, and therefore do not explicitly model wave effects. Without wind-wave coupling, such simulations cannot provide information regarding the sea state in the hurricane. Such coupling is needed to explore how the hurricane atmospheric and oceanic boundary layers would affect the entire turbine and support
structures such as monopiles, jackets and floating platforms and mooring systems. The work presented herein has demonstrated that the hurricane boundary layer can adversely affect offshore wind turbines, and motivates subsequent examination with further refinements.

## 6 Acknowledgements

The authors acknowledge the financial support of the US National Science Foundation through grants CMMI-
1552559 and CMMI- 1234656 and the University of Bologna through grant Prot. N.1067 tit. III cl.12. They also acknowledge high‐performance computing support from Yellowstone (ark:/85065/d7wd3xhc) provided by the National Center for Atmospheric Research's Computational and Information Systems Laboratory and sponsored by the National Science Foundation. This work was authored, in part, by the National Renewable Energy Laboratory, operated by Alliance for Sustainable Energy, LLC, for the U.S. Department of Energy (DOE) under Contract No.
DE-AC36-08GO28308. Funding provided by the U.S. Department of Energy Office of Energy Efficiency and Renewable Energy Wind Energy Technologies Office. The views expressed in the article do not necessarily represent the views of the DOE or the U.S. Government. The U.S. Government retains and the publisher, by accepting the 5 article for publication, acknowledges that the U.S. Government retains a nonexclusive, paid-up, irrevocable, worldwide license to publish or reproduce the published form of this work, or allow others to do so, for
U.S. Government purposes.

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
