# Peer review of "Hurricane eyewall winds and structural response of wind turbines"

_Wind Energy Science, 2019_

## Referee Comment (RC1) · Anonymous Referee #1 · 1 May 2019

General Comments This paper presents the use of a high-fidelity wind data from an LES simulation of a Cat 5 hurricane as wind input to drive the aero-elastic loading of the DTU 10-MW turbine to show the impact of hurricane winds on structural loading. The paper is very well written and the topic of relevance to the offshore wind turbine industry, but there are some issues that need to be addressed before final publication.

Specific Comments - Page / Line / Comment 4 / 115 / Why was the analysis of the LES data for use in TurbSim limited to shear, veer, and TI? The TurbSim tool can generate synthetic wind from more information than is used here. It appears possible to derive the turbulence spectra in u,v,w, the spatial coherence in u,v,w, and the Reynold's stresses (vector component correlations) from the LES data. 10 / 211 / The previous section focused a lot on gust factors and direction changes. How do these compare

between the LES data and the resulting TurbSim simulations? 10 / 217 / The wind data extracted from the LES was only 10-minutes long. Is it OK to assume that the same 10-minutes of wind conditions would apply for a full 1-hour? Wouldn't increasing the length of the averaging period result in a reduction of storm severity e.g. a reduction in mean wind speed? 11 / 233 / This section does not mention how the transverse turbulence is considered. How do the v and w components of the turbulence compare? 11 / 239 / From the understanding of the reviewer, Figure 4 shows the maximum instantaneous veer over the 10-minute LES data. Why is this veer used as the 1-hour average veer in TurbSim / FAST? Shouldn't the 10-minute average veer be used? It appears that the veer applied here is far stronger than what is physical. 12 / 249 / Can't the turbulence spectra be derived directly from the LES? Why not simply use that directly in TurbSim? Why introduce an error by using a Kaimal spectrum with only the TI matching LES? Also, what about the velocity spectra for the v and w components? 14 / 260 / What values of "a" and "b" where used? Wouldn't it be better to fit "a", "b", and "CohExp" collectively based on the LES data? And why not do this separately for each case? 15 / 281 / What features in FAST are enabled? Has induction been disabled for the hurricane / idling cases? Are aerodynamic loads on the tower considered? Is BeamDyn enabled to permit the modeling of large blade deflections? 15 / 283 / Why compare the hurricane loads to loads while the turbine is operating at rated? Wouldn't it be better to compare the hurricane loads directly against what a standard IEC-based design would consider e.g., load cases 6.1 and 6.2 based on class I winds and category "A" turbulence? 15 / 289 / It is unclear to the reviewer how the yaw misaligments were derived. Are you assuming that the yaw controller is always active? Are you assuming the time it takes for the yaw controller to measure the yaw misalignment before inducing yaw motion? 16 / 298 / Modeling high wind events with yaw errors in the range of 20-30 degrees typically result in aero-elastic instability (negative damping) of the blade-edgewise mode, which results in unrealistically large blade deflections and tower side-to-side loads. Do you see any signs of this instability here? The wind industry typically neglects yaw errors that result in this instability because they don't believe

the numerical models accurate predict deflections and loads under these conditions. 17 / 311 / A tower has a cylindrical cross section, so, the maximum loads should be assessed in terms of the vector magnitude / resultant, rather than specific fore-aft and side-to-side components. The reviewer suggests eliminating the rows associated with FA and SS. Also, why are the blade-root loads not shown? These too should be shown in terms of the vector magnitude, not the vector component. 17 / 316 / The text here and later refers to "blade loads", but only "blade deflections are shown. The reviewer suggests adding blade loads to these tables. 17 / 330 / It is unclear where the 5.4 10ˆ5 kNm load and 18-m deflection limit come from. Please clarify. 19 / 375 / Such large deflections cannot reasonably be modeled with FAST's ElastoDyn module. Was BeamDyn applied here? 19 / 380 / Is this blade root moment the vector magnitude / resultant, or something else? (It should be the resultant because the blade root is an axisymmetric structure.)

Technical Corrections - Page / Line / Comment 1 / 36 / The reviewer hasn't checked all of the references, but this publication by Tarp-Johansen et al does not appear in the Reference section. 3 / 86 / The word "fully" has no technical meaning. Suggest eliminating the word "fully". 3 / 94 / This paper does not discuss the probability of occurrence of the events simulated. It may be that it is not wise to design the structure to withstand the loads under the events simulated if the probability is very low. It may be better to accept this risk and insure for it. Some mention of this is likely worth it in the paper introduction. 7 / 159 / If important, a battery back-up system could be installed to ensure that yaw control is not lost even if the grid is lost. It may be worth mentioning this here, but it doesn't appear that grid loss was assumed in the further FAST analysis. 10 / 222 / You reiterate that the nonGaussian wind is not simulated in TurbSim. It may be worth referencing here work from others that has shown that assuming Gaussian wind tends to be a reasonable assumption without being nonconservative in regards to loads prediction. 11 / 233 / It appears that TurbSim always simulated a lower TI than prescribed by LES. This underprediction can be eliminated by using the ScaleIEC option in TurbSim. 18 / 336 / Change "wind speed profile veer" to "wind speed profile". 20 / 405 / Earlier in the paper you say that the Cat 5 simulation is not a severe Cat 5 hurricane, but now you call this "a worst-case scenario"? Choose your wording carefully.

---

## Referee Comment (RC2) · Anonymous Referee #2 · 4 Jun 2019

One major comment I have with this version of the manuscript is the representation of a hurricane in a wind field generator such as FAST.

In general we need to generate many wind seeds to represent the stochastic nature of the wind when determining the maximum loading on the blade or on the tower. How many stochastic simulation have been carried out in FAST to represent this stochastic variation. In this case the maximum loads should be represented by the statistics, whether as the median value or as percentile.

When transferring the information of the simulated hurricane in LES to a simplified representation such as TurbSim, the non-stationary nature of the hurricane wind and coherence of the hurricane structure is lost. Furthermore, how many times were the hurricane wind simulated. If it were simulated only one time, then the statistical com-

parison is not really valid, as through more simulations the statistics will change as well. In order to make meaningful statistics and statements on the increased loading on the structure, it is necessary to run several simulations in order to capture the statistical variation of the inflow conditions and of the response of the wind turbine.

How accurate is the representation of the hurricane using LES, especiall the convective flow within the hurricane driven by the temperature gradients, sea surface temperature and air temperature. the vertical component of the wind speed may play a significant role for the loading, how well is this vertical component of the wind velocity captured by the model.

some minor comments can be found in the attached PDF file

Please also note the supplement to this comment:
https://www.wind-energ-sci-discuss.net/wes-2019-14/wes-2019-14-RC2-supplement.pdf

**Supplement:**

[revised manuscript text omitted]

*the LES simulation is stochastic so the statistics from the LES simulation data will differ from realization to realization.*

[Figure]

[Figure]

[Figure]

**Figure 5: Wind shear profile: the mean wind shear profile at R = 10, 12, 15 and 20 km for both the LES and TurbSim wind field simulations. For reference, the three horizontal lines indicate the top and bottom of the rotor disk, at 208 m and 30 m, respectively and the hub height at 119 m.**

245     The power spectral density compares well between LES and TurbSim (Fig. 6). The hub-height wind speed power spectral density created by TurbSim is modeled with a Kaimal spectrum. The temporal resolution of the LES data and the implicit subgrid-scale filter causes the power to fall off at high frequencies, more quickly than the TurbSim data. Aside from this expected difference, the plots show a reasonable agreement between the two datasets at all four radii.

what about the parametrization of the LES simulation. Which are the parameter that can affect the comparison with TurbSim and how meaningful is the comparison?

[revised manuscript text omitted]

---

## Author Comment (AC1) · 18 Sep 2019

In response to two reviews we have made several corrections and added explanation to several points. Most significantly, we have re-analyzed our simulation data as a series of six ten-minute intervals rather than a single one-hour interval to allow for uncertainty between simulations of a stochastic wind field. This re-analysis resulted in only very minor changes to the numerical results and no change to the qualitative interpretation of those results. Nevertheless, we agree with the reviewer that presenting maxima from a single one-hour simulation improperly neglected stochasticity of the response.

We have prepared and uploaded a detailed response to reviewers and have typed any new text in the manuscript in bold face so that the reviewers and editors may more

easily identify changes.

We appreciate the time and effort devoted to the review process.

Please also note the supplement to this comment:
https://www.wind-energ-sci-discuss.net/wes-2019-14/wes-2019-14-AC1-
supplement.zip

---

## Author Response (AR1)

**Reviewer 1:**

**General Comments** This paper presents the use of a high-fidelity wind data from an LES simulation of a Cat 5 hurricane as wind input to drive the aero-elastic loading of the DTU 10-MW turbine to show the impact of hurricane winds on structural loading. The paper is very well written and the topic of relevance to the offshore wind turbine industry, but there are some issues that need to be addressed before final publication.

**Specific Comments - Page / Line / Comment**:
1. 4 / 115 / Why was the analysis of the LES data for use in TurbSim limited to shear, veer, and TI? The TurbSim tool can gen- erate synthetic wind from more information than is used here. It appears possible to derive the turbulence spectra in u,v,w, the spatial coherence in u,v,w, and the Reynold's stresses (vector component correlations) from the LES data.

Our objective in choosing shear, veer and TI, along with direction change, was to restrict ourselves to the study of the role of the primary features of interest of the LES wind field that occur on the scale of the turbine structure and rotor. Indeed Turbsim could have generated wind fields using more input from LES, but we felt doing so would have risked make the results overly storm-specific and have made the results less readily interpretable in terms of gross features such as shear, veer, TI and direction change. Similarly, while including separate wind characteristics in the v and w directions would have been possible we concluded that it would have overly expanded the parameter space.

We have added some text to the paper to clarify our intentions in this regard and acknowledge the possibility of more completely replicating the LES wind characteristics.

**The remainder of the paper focusses on the effect of mean wind speed, veer, and direction change / misalignment on turbine loads. These characteristics have been selected from the richer set of parameters available from the LES simulations to most closely mimic, in the authors' experience, the parameters most often treated as primary in typical design-level analysis.**

2. 10 / 211 / The previous section focused a lot on gust factors and direction changes. How do these compare between the LES data and the resulting TurbSim simulations?

This is a good valid question. Regarding direction change, there is no mean direction change in TurbSim so we used yaw misalignment as a surrogate.

Returning to our TurbSim data we find a gust factor of approximately 1.3, significantly lower than the maxima detected in the LES data. The reviewer is correct that we spend a fair amount of space describing the LES gusts, but in the remainder of the paper focus on mean wind speed, shear, coherence, veer and direction change. Therefore we have added some text to describe the shortcoming in TurbSim's modeling of the gust factors and to urge further study in how TurbSim may be able to generate wind files with more extreme gust factors.

**Using the inputs as described TurbSim also did not replicate the gust factors observed in the LES data, with the TurbSim gust factors reaching a maximum of 1.3, significantly lower than observed in the LES data. Generating more intense gusts in Turbsim would require increasing the turbulence intensity or modifying the distribution of the wind speed and such approaches should be developed in further work.**

3. 10 / 217 / The wind data extracted from the LES was only 10-minutes long. Is it OK to assume that the same 10-minutes of wind conditions would apply for a full 1-hour? Wouldn't increasing the length of the averaging period result in a reduction of storm severity e.g. a reduction in mean wind speed?

We believe the assumption of stationarity is valid for a 1-hour period during a storm based on our experience with hurricane data and simulations. It is correct that examining a longer simulation record would result in a change in some statistics, but the key statistics used here, mean wind speed and TI, are relatively stable between 10 min and 1 hour periods. Furthermore, we face the practical limitation that generating the 10 minutes of data required extensive supercomputer time and that extending the simulation periods was simply not possible for this study.

4. 11 / 233 / This section does not mention how the transverse tur- bulence is considered. How do the v and w components of the turbulence compare?

The reviewer makes a good point. We have indeed assumed that the transverse components have identical TI in the name of simplicity of the modeling and because in practice we are not aware that designers and analysts typically prescribe different TIs in the different directions. We have added some text to the MS to explain our reasoning in this regard.

**Note that identical turbulence intensity has been applied to all three components of the wind field. To the author's knowledge this is common practice in design and analysis of offshore wind turbines. Further study of the possibility of turbulence intensity differences in the component wind directions would be a worthwhile further contribution.**

5. 11 / 239 / From the understanding of the reviewer, Figure 4 shows the maximum in- stantaneous veer over the 10-minute LES data. Why is this veer used as the 1-hour average veer in TurbSim / FAST? Shouldn't the 10-minute average veer be used? It appears that the veer applied here is far stronger than what is physical.

Our intention was to look at a worst case veer situation. Averaging veer over 10 minutes is likely to overly smooth the veer towards a zero veer case. The reviewer is correct that consideration of the correct averaging period for veer load cases is an interesting and important topic and we will consider pursuing this further as part of other work but as of now no standard approach exists for characterizing veer for OWT design and we therefore selected a worst case example.

We have added some text to clarify the choice of instantaneous veer.

**It should be noted that the veer profiles selected are worst case instantaneous veer profiles from the 10-minute simulation. Instantaneous veer profiles may overestimate the veer that drives structural response due to time scales associated with structural response to direction change. In the absence of consensus on how to average veer profiles to establish design veer profiles, the instantaneous worst case has been used here with the caveat that result may overestimate the impact of veer on structural response.**

6. 12 / 249 / Can't the turbulence spectra be derived directly from the LES? Why not simply use that di- rectly in TurbSim? Why introduce an error by using a Kaimal spectrum with only the TI matching LES? Also, what about the velocity spectra for the v and w components?

With the 10 minutes available, the high frequency noise in the spectrum and the high frequency roll-off we did not feel confident that we could calibrate a complete user-defined spectrum from the LES data. The high frequency roll-off observed in the LES spectra is likely also a function of temporal discretization and therefore should not be included in the FAST/TurbSim wind field.

Furthermore, as with the turbulence intensities, we made a decision to manage the number of parameters in a way that is consistent with our understanding of how FAST is typically used in design and analysis. This means using a well established spectral model such as Kaimal rather than trying to fit the LES spectra exactly and assuming the same spectral form for the transverse v and w components. Certainly it would be possible to more completely capture the LES wind characteristics in FAST/TurbSim and we have noted this in text added to the MS:

**While it is possible to specify a user-defined spectrum in TurbSim, the Kaimal spectrum was selected as a model to not make the simulations overly storm-specific, to mitigate effects of temporal resolution in creating artifacts such as high frequency roll-off in the spectra and to mimic what the authors understand to be standard practice in design and analysis. Furthermore, as with the turbulence intensities, transverse components of the wind field have been assumed to have the same spectral form.**

7. 14 / 260 / What values of "a" and "b" where used? Wouldn't it be better to fit "a", "b", and "CohExp" collectively based on the LES data? And why not do this separately for each case?

We have added some text explaining our reasoning for using CohExp as the fitting parameter.

**The *CohExp* parameter provides primary control in the TurbSim input files and has been selected as the only coherence fitting parameter in the name of simplicity.**

8. 15 / 281 / What features in FAST are enabled? Has induction been disabled for the hurricane / idling cases? Are aerodynamic loads on the tower considered? Is BeamDyn enabled to permit the modeling of large blade deflections?

Induction has been disabled, tower aerodynamic loads are included and BeamDyn was not enabled. We have clarified this in the MS and address the issues with large beam deflections elsewhere in the response.

**induction is disabled and aerodynamic loads on the tower are included.**

9. 15 / 283 / Why compare the hurricane loads to loads while the turbine is operating at rated? Wouldn't it be better to compare the hurricane loads directly against what a standard IEC-based design would consider e.g., load cases 6.1 and 6.2 based on class I winds and cate- gory "A" turbulence?

We included the rated case because it provides a reference case for judging the magnitude of structural response and loading. We wanted to avoid comparison to specified IEC load cases since the Cat 5 storm studied is beyond the range of what will likely every be included in design specifications and we wanted to avoid the misconception that we were proposing the category 5 storm as a new design standard.

The rated case also has the secondary advantage of being a non-site-specific load case.

10. 15 / 289 / It is unclear to the reviewer how the yaw misaligments were derived. Are you assuming that the yaw controller is always active? Are you assuming the time it takes for the yaw controller to measure the yaw misalignment be- fore inducing yaw motion?

Yaw misalignments are used as a surrogate for rapid direction changes in the hub height wind direction. Since the hub height wind direction cannot be changed temporally in FAST/TurbSim we used a consistent yaw misalignment to gauge the possible effect of rapid direction change. Applied yaw misalignments were chosen to reflect extreme 10s and 30s direction changes observed in the LES wind field. An underlying assumption is that some of these direction changes occur so rapidly that even a responsive yaw-controller could not react quickly enough.

We have attempted to clarify this in the MS:

**Since mean wind direction cannot vary temporally in FAST/TurbSim yaw misalignment has been used as a surrogate for the effects of rapid direction change.**

11. 16 / 298 / Modeling high wind events with yaw errors in the range of 20-30 degrees typically result in aero-elastic instability (negative damping) of the blade-edgewise mode, which results in unrealistically large blade deflections and tower side-to-side loads. Do you see any signs of this instability here? The wind indus- try typically neglects yaw errors that result in this instability because they don't believe the numerical models accurate predict deflections and loads under these conditions.

We did not observe signs of these instabilities, which, from internal experience in our research group, do not appear as significant an issue with the DTU 10MW model as was the case with the NREL 5MW model.

12. 17 / 311 / A tower has a cylindrical cross section, so, the maximum loads should be assessed in terms of the vector magnitude / resultant, rather than specific fore-aft and side-to-side components. The reviewer suggests eliminating the rows associated with FA and SS. Also, why are the blade-root loads not shown? These too should be shown in terms of the vector magnitude, not the vector component.

The tower base resultant moment is included in the tables.  We choose to retain the FA and SS components in the results tables because they do help interpretation of the effects of the hurricane wind fields on motion of the tower in the two principal directions.

In response to the reviewer's comment we have added blade root moments to the results tables.

13. 17 / 316 / The text here and later refers to "blade loads", but only "blade deflections are shown. The reviewer suggests adding blade loads to these tables.

Blade root moments have been added to the results tables.

14. 17 / 330 / It is unclear where the 5.4 10^5 kNm load and 18-m deflection limit come from. Please clarify.

We have added the following clarifying text and references

**The design of the DTU 10 MW turbine is specified by Bak et al. (2013). The base cross-section of the tower is a hollow steel tube with a diameter of 8.3 m and a thickness of 38 mm. For a section with this geometry and with Imperfection Quality Class B, Eurocode (EN 1993 1-6) prescribes a flexural strength of 74% of the yield moment. For the DTU 10 MW turbine tower, which is made with S355 steel, which has a yield stress of 355 MPa, the flexural strength of this section per Eurocode is is 5.4e5 kN-m. The 18 m blade deflection limit is a rounded limit based on the 18.3 m tower clearance in the DTU 10 MW turbine (Bak et al. 2013).**

15. 19 / 375 / Such large deflections cannot reasonably be modeled with FAST's ElastoDyn module. Was BeamDyn applied here?

We did not apply BeamDyn and acknowledge the elastodyn is not able to correctly model the very large displacements observed.  That is why we had tried to emphasize that the large displacement valued were not to be taken as quantitatively meaningful, but rather as an indication that large blade deflections of potential structural concern were occurring.  We have tried to clarify this with additional language in the MS:

**This is a physically unrealistic tip deflection, beyond the modeling and simulation capabilities of FAST / ElastoDyn as used in this study.  The numerical magnitudes of the tip deflections, therefore, should not be taken as quantitatively meaningful, but rather as an indication that large blade deflections, of potential structural concern, are occurring during the loading.**

16. 19 / 380 / Is this blade root moment the vector magnitude / resultant, or something else? (It should be the resultant because the blade root is an axisymmetric structure.)

Yes, it is resultant as indicated in the figure caption.

**Technical Corrections - Page / Line / Comment**:
1. 1 / 36 / The reviewer hasn't checked all of the references, but this publication by Tarp-Johansen et al does not appear in the Reference section.

We have added the reference.

2. 3 / 86 / The word "fully" has no technical meaning. Suggest eliminating the word "fully".

We have deleted the word 'fully'

3. 3 / 94 / This paper does not discuss the probability of oc- currence of the events simulated. It may be that it is not wise to design the structure to withstand the loads under the events simulated if the probability is very low. It may be better to accept this risk and insure for it. Some mention of this is likely worth it in the paper introduction.

This is a good point and one that should have been given more attention in the paper. Additional explanation of this matter has been included in the revision.

**The category 5 storm simulation used to develop the wind fields used in this paper is a severe storm, stronger than any likely to directly affect the US Atlantic coast from the Carolinas northward. Therefore, it is emphasized that the primary purpose of this paper is not to establish that a category 5 hurricane imposes large loads on offshore wind structures, but rather that hurricanes contain wind field characteristics that are not currently considered in design and that may exacerbate loading in unexpected ways. In fact, it is unlikely that design codes should require resistance to a category 5 storm since such a storm is so unlikely for most proposed wind energy areas.**

4. 7 / 159 / If important, a battery back-up system could be installed to ensure that yaw control is not lost even if the grid is lost. It may be worth mentioning this here, but it doesn't appear that grid loss was assumed in the further FAST analysis.

Our purpose was to use yaw misalignment as a surrogate for very rapid (10s- 30s) direction changes. In such cases the turbine will be subject to non-aligned wind even in the case of a functioning, battery-supported, yaw control system.

5. 10 / 222 / You reiterate that the nonGaussian wind is not simulated in TurbSim. It may be worth referencing here work from others that has shown that assuming Gaussian wind tends to be a reasonable assumption without being nonconservative in regards to loads prediction.

We appreciate the reviewers comment. If there are specific such references the reviewer has in mind we would be pleased to include them.

6. 11 / 233 / It appears that TurbSim always simulated a lower TI than prescribed by LES. This underprediction can be eliminated by using the ScaleIEC option in TurbSim.

We appreciate the advice from the reviewer and will investigate and apply this option in further studies.

7. 18 / 336 / Change "wind speed profile veer" to "wind speed profile".

We have mode this change.

8. 20 / 405 / Earlier in the paper you say that the Cat 5 simulation is not a severe Cat 5 hurricane, but now you call this "a worst-case scenario"? Choose your wording carefully.

We thank the reviewer for noting this inconsistency. Indeed we did refer to the simulated storm as both 'relatively small' and 'worst-case'. We have rephrased to fix this seeming contradiction.

**This Category 5 storm, even though small in size compared to many category 5 storms, represents an environmental scenario far more severe than would credibly affect the vast majority of wind energy areas.**

**Reviewer 2:**

**General comments:** One major comment I have with this version of the manuscript is the representation of a hurricane in a wind field generator such as FAST.
In general we need to generate many wind seeds to represent the stochastic nature of the wind when determining the maximum loading on the blade or on the tower. How many stochastic simulation have been carried out in FAST to represent this stochastic variation. In this case the maximum loads should be represented by the statistics, whether as the median value or as percentile.
When transferring the information of the simulated hurricane in LES to a simplified representation such as TurbSim, the non-stationary nature of the hurricane wind and coherence of the hurricane structure is lost. Furthermore, how many times were the hurricane wind simulated. If it were simulated only one time, then the statistical comparison is not really valid, as through more simulations the statistics will change as well. In order to make meaningful statistics and statements on the increased loading on the structure, it is necessary to run several simulations in order to capture the statistical variation of the inflow conditions and of the response of the wind turbine.
How accurate is the representation of the hurricane using LES, especiall the convective flow within the hurricane driven by the temperature gradients, sea surface temperature and air temperature. the vertical component of the wind speed may play a significant role for the loading, how well is this vertical component of the wind velocity captured by the model.
some minor comments can be found in the attached PDF file

We thank the reviewer for the comments regarding the stochasticity of the wind field. Indeed the original version of the manuscript included maximum load and displacement effects from a single 1-hour simulation of the turbine response.

We have reanalyzed the data by dividing the initial 1 hour simulation into 6 ten minute time intervals, computing the maximum response during each interval and averaging those maxima. This has the additional advantage that the ten minute interval corresponds to the duration of the LES simulation. We found that the results as tabulated change only very modestly and that no qualitative changes in the interpretation were required by this alteration of the analysis method.

**A one-hour simulation of the turbine response was performed for each wind field case. This record was then subdivided into six ten-minute intervals and the maximum turbine response was extracted for each ten-minute interval. Table 3 summarizes the wind field characteristics for the full set of simulation cases, and the following table provide the average of the 6 maximum responses of the ten-minute simulation intervals.**

Details regarding the LES modeling and accuracy are provided at length in the referenced articles:

Bryan, G. H., Worsnop, R. P., Lundquist, J. K. and Zhang, J. A.: A Simple Method for Simulating Wind Profiles in the Boundary Layer of Tropical Cyclones, Bound.-Layer Meteorol., doi:10.1007/s10546-016-0207-0, 2016.

Worsnop, R. P., Lundquist, J. K., Bryan, G. H., Damiani, R. and Musial, W.: Gusts and shear within hurricane eyewalls can exceed offshore wind turbine design standards, Geophys. Res. Lett., 44(12), 2017GL073537, doi:10.1002/2017GL073537, 2017a.

Worsnop, R. P., Bryan, G. H., Lundquist, J. K. and Zhang, J. A.: Using Large-Eddy Simulations to Define Spectral and Coherence Characteristics of the Hurricane Boundary Layer for Wind-Energy Applications, Bound.-Layer Meteorol., 1–32, doi:10.1007/s10546-017-0266-x, 2017b.

We have decided to omit these details from this paper that is focused on the engineering performance of the wind turbine.

**Specific comments:**
1. Page 11: The LES simulation is stochastic so the statistics from the LES simulation data will differ from realization to realization.

We have attempted to address this by the reanalysis described above.

2. Page 12: What about the parametrization of the LES simulation? Which are the parameters than can affect the comparison with Turbsim and how meaningful is the comparison?

We have parametrized the LES wind fields in terms of their mean wind speed, turbulence intensity, spectrum, coherence, shear and veer. We believe this is fully described in the paper but welcome suggestions for clarification.

3. Page 17: Why the blade tip deflection in out of plane direction in table 6 so low compared to in plane direction?

We believe this is due to the influence of VEER. With the blades feathered they present a slim profile to the predominant wind direction, but VEER causes localized wind to act flapwise (in plane) to the blade.

4. Page 18: The out of plane deflection is very small compared to in plane in table 8

The explanation is similar to that for Table 6.

---

## Author Response (AR2)

College of Engineering

**Department of Civil & Environmental Engineering**

Sanjay R. Arwade, Professor

arwade@umass.edu, 413 577 0926

October 31, 2019

Dear Prof. Meyers:

Following are our responses to the minor revisions requested by the two reviewers. Thank you again for the time and effort on your part as well as on the part of the WES staff and the reviewers.

Sincerely,

Sanjay R. Arwade
Professor

**Reviewer 1:**

We thank the reviewer for the pointer to the preferred reference on non-Gaussian winds and have included the following reference:

Schoettler, J., Reinke, N., Hölling, A., Whale, J., Peinke, J. and Hölling, M.: On the impact of non-Gaussian wind statistics on wind turbines – an experimental approach, Wind Energ. Sci., 2, 1-13 doi: https://doi.org/10.5194/wes-2-1-2017, 2017

**Reviewer 2:**

**Comment:** In the manuscript it is mentioned (page 15) that

"This record was then subdivided into six ten-minute intervals and the maximum turbine response was extracted for each ten-minute interval. Table 3 summarizes the wind field characteristics for the full set of simulation cases, and the following table provide the average of the 6 maximum responses of the ten-minute simulation intervals"

what is the reason behind subdividing the 1 hour simulation. Since the one hour simulation is produced with the same random seed, there are some correlation between 10 minutes blocks. Therefore the average of the extreme responses of the six blocks is affected. Possibly the average is not affected significantly but the standard deviation would be larger if these blocks were statistically independent.

**Response:** The reviewer is correct that the ten-minute simulations are not independent.  However, due to the short time scales associated with the turbulence, and the absence of longer period wave loading, the effect of this correlation is negligible and outweighed by the advantage of providing some sense of the stochastic nature of the statistics.  We have added an acknowledgement of this lack of independence:

"Subdividing a one-hour simulation into ten-minute intervals introduces a minor correlation between the ten-minute intervals, but since the time scales associated with turbulence are very small compared to the ten-minute analysis interval this effect is negligible.  This procedure mitigates the need to allow transients to dissipate for each ten-minute simulation and provides the ability to estimate variability across simulations."